# Review of Novel Oral Amphotericin B Formulations for the Treatment of Parasitic Infections

**DOI:** 10.3390/pharmaceutics14112316

**Published:** 2022-10-28

**Authors:** Ellen Wasan, Tavonga Mandava, Pablo Crespo-Moran, Adrienne Nagy, Kishor M. Wasan

**Affiliations:** 1College of Pharmacy and Nutrition, University of Saskatchewan, Health Sciences Building, Saskatoon, SK S7N 5E5, Canada; 2Department of Urologic Sciences, Faculty of Medicine & the Neglected Global Diseases Initiative, University of British Columbia, Vancouver Campus, Vancouver, BC V5Z 1L8, Canada

**Keywords:** oral amphotericin B, safety and tolerability, pharmacokinetics, parasitic infections, human use, veterinary use, nanomedicines, drug delivery, nanoparticles, SEDDS

## Abstract

Amphotericin B (AmpB) is a polyene macrolide antibiotic used in the treatment of blood-borne parasitic and fungal infections. However, its use, particularly in the developing world, has been limited by dose-dependent kidney toxicity, other systemic-related toxicity issues following injection, the inconvenience of parenteral administration, and accessibility. Oral formulation approaches have focused on the dual problem of solubility and permeability of AmpB, which is poorly water soluble, amphoteric and has extremely low oral bioavailability. Therefore, to enhance oral absorption, researchers have employed micellar formulations, polymeric nanoparticles, cochleates, pro-drugs, and self-emulsifying drug delivery systems (SEDDS). This paper will highlight current uses of AmpB against parasitic infections such as leishmaniasis, preclinical and clinical formulation strategies, applications in veterinary medicine and the importance of developing a cost-effective and safe oral AmpB formulation.

## 1. Introduction

Protozoa of the *Leishmania* genus are obligate intracellular parasites spread by the bite of a sandfly, with a life cycle that includes promastigote residence in macrophages in addition to amastigote spread into tissues (Figure 1) [1]. There are more than 20 species that infect humans and animals. Leishmaniasis has multiple forms: visceral, cutaneous and mucocutaneous. Visceral leishmaniasis attacks internal organs such as the spleen, liver and bone marrow, whereas the cutaneous forms affect skin and skin structures which can cause major disfigurement and disability. In 2022, The World Health Organization (WHO) stated that visceral leishmaniasis (VL, “kala-azar”) is fatal if left untreated in over 95% of cases. It remains one of the top parasitic diseases with outbreak and mortality potential. Approximately 50,000 to 90,000 annual new cases of VL occur globally, and it is estimated that only a quarter of cases may be reported to WHO [2]. In 2020, more than 90% of new cases reported to WHO occurred in 10 countries: Brazil, China, Ethiopia, Eritrea, India, Kenya, Somalia, South Sudan, Sudan and Yemen. Hundreds of millions of people are at risk in endemic areas. The most recent estimates from the Global Burden of Disease Study (GBD) [3] indicate that leishmaniasis causes > 24,000 deaths annually and 3.3 million disability-adjusted life years (DALYs). Thus, leishmaniasis represents a neglected tropical disease (NTDs) with an extremely high disease burden [4]. While further developing vaccines and insect control measures are essential for managing leishmaniasis, effective drug therapy regimens are still needed.

## 2. Amphotericin B and Formulations in Clinical Use

Based on its effectiveness, Amphotericin B (AmpB) is the drug of choice for VL, preferably in combination therapy, which may include paramomycin (an aminoglycoside) or miltefosine (contraindicated in pregnancy) [5]. In some regions, pentavalent antimonial (Sb^V^) is used, although resistance is noted in some geographical regions. Special consideration of dosing regimens is taken for immunocompromised patients such as those with concomitant HIV infection [6]. AmpB, a polyene macrolide antibiotic, is used in the treatment of both serious fungal and parasitic infections. (Table 1) While highly effective, its use has been limited by dose-dependent renal toxicity and toxicities associated with its parenteral administration. These adverse effects include anemia, hypomagnesemia, fever, chills, rigors, abdominal pain and gastrointestinal symptoms, increases in blood urea nitrogen and creatinine and many others [7]. Lipid-based formulations of AmpB are available and widely used to reduce these toxicities, however, they are limited by a lack of cost-effectiveness related to parenteral administration and the requirement for reliable cold-chain shipping and storage. The mechanism of action of AmpB involves its ability to bind to the sterol component of fungal and/or parasitic cell membranes causing cell lysis and ultimately cell death. Additional models of its mechanism have recently expanded to include surface adsorption and sterol sponge effects which disrupt the fungal or parasitic membranes [8]. Its uses include serious, life-threatening systemic fungal, cryptococcal and parasitic infections, indicated in Table 1 [9].

A significant challenge with AmpB as a drug compound is its poor solubility and low permeability across gastrointestinal membranes. AmpB (Figure 2, chemical structure) has a water solubility of only 0.1 mg/mL in water at pH2 or 11 and is insoluble at neutral pH [10]. Its logP is 0.8 and k_ow_ = −2.8 and its pKa values are 3.5 and 9.11. These features, along with its high molecular weight of 924 Da and negligible gastrointestinal or blood–brain barrier permeability do not make AmpB a typical druggable molecule, placing it in BCS Class IV.

AmpB is formulated for parenteral administration either into micellar suspension with sodium deoxycholate, or into various lipid-based drug delivery systems using phospholipids (Table 2) [11,12,13,14,15]. Compared to the micellar formulation of AmpB (marketed as Fungizone^®^ in North America), the lipid-based formulations have been shown to reduce acute toxicity. The micellar AmpB formulation is currently administered as a slow infusion which can result in immediate adverse effects, including fever, chills, rigors, nausea, vomiting, hyperpyrexia, severe malaise, hypotension, thrombophlebitis, cardiac enlargement, anaemia, and hepatitis. These reactions appear most commonly in the first week of administration and vary considerably between patients, which may be dose-limiting. Adverse effects may diminish as therapy progresses, and pre-medications are used to reduce symptoms, however, in some patients, therapy must be discontinued, which is disadvantageous for treatment of the infection. Nephrotoxicity is the most common acute adverse reaction following micellar AmpB administration as indicated by rising serum creatinine and urea levels and is often accompanied by hypokalemia. Renal function may normalize on discontinuation of micellar AmpB, but in some patients, irreversible damage may occur. Lipid complex and liposomal AmpB were developed to modify the biodistribution of the drug and to enhance efficacy. Various forms of liposomal AmpB are now manufactured globally and are the preferred form where available. Liposomal AmpB has less of the acute systemic and renal toxicity of the micellar form and therefore far less therapy-limiting adverse effects. The major limitation to liposomal AmpB, however, is not adverse effects but cost and accessibility which impacts rural and economically disadvantaged areas the most [5,16]. Dosage forms that enable oral dosing would not only be cost effective but would expand AmpB’s utility. This review provides a current view on a variety of formulation designs to enhance the solubility and/or bioavailability of AmpB, particularly by the oral route. To overcome the limitations of parenteral AmpB formulations, the development of an oral formulation of AmpB that is cost-effective, accessible, less-toxic yet equally efficacious would be ideal [17,18,19,20].

## 3. Marketed Formulations of AmpB

AmpB has been used extensively as an antiparasitic therapy for decades; however, in light of its associated nephrotoxicity [21], various formulations have been developed to reduce its inherent toxicity. Table 2 shows Food and Drug Administration (FDA) approved AmpB formulations. Among these formulations the most effective and most extensively studied is AmBisome^®^ which is considered to be the least nephrotoxic and most efficacious of all. The lipid-based formulations are administered via the intravenous route, and as noted above, the side effects and high cost significantly limit use in developing countries [22]. The approved commercial formulations use lipids as an effective drug delivery system, not only to solubilize the drug but also to modulate the pharmacokinetics and tissue distribution, and this approach is still under investigation [23]. AmBisome^®^ is comprised of cholesterol-containing small unilamellar vesicles with AmpB embedded into the liposomal membrane. AmpB lipid complex (ABLC; Abelcet^®^) forms ribbon-like drug-phospholipid aggregates, whereas AmpB colloidal dispersion (ABCD; Amphotec^®^) consists of 1:1 AmpB-cholesteryl sulphate complex [13]. More recently, a mixture of AmB-deoxycholate complex with a lipid emulsion (ABLE) was developed and licensed in India [24].

## 4. Novel AmpB Parenteral Formulations in Development

As noted in Table 3, several new delivery strategies have been developed for AmpB during the last decade. One of the most broadly studied drug delivery systems for AmpB is encapsulation into liposomes [22,25,26,27,28] as the safest and most efficient formulation achieved to date. However, liposomal formulations are relatively expensive and retain some degree of toxicity, and so others have begun introducing nanoparticles and other lipidic structures such as emulsions as possible alternatives. Polymer and protein-based nanoparticles also show promise in terms of toxicity and efficiency. Polymeric formulations have generally increased the biocompatibility features of AmpB, while proteins allow the specific targeting of bodily structures. Prasanna et al. [27] also describe solid lipid nanoparticles (SLNs) of ~ 50–1000 nm, with a solid lipid matrix stabilized by physiologically compatible emulsifiers.

### 4.1. Macrophage-Targeted Formulations

As indicated in Figure 1, part of the leishmania parasite’s life cycle is spent in phagocytic macrophages, which may not be accessible to conventional drug therapy. Specific strategies are needed to address this pool of residual disease; if the host becomes immune compromised, reinfection from this reservoir is possible. Table 4 compiles the AmpB formulations that are exploring the use of mannose as a targeting agent to reach macrophages directly. Solid lipid nanoparticles are one of the techniques used to target macrophages as the drug release site [42]. Strategies that incorporate mannose into nanoparticles to take advantage of mannose receptors on macrophages, allowing for better specificity of delivery. SLNs designed for this purpose must exhibit low cytotoxicity towards macrophages, a high degree of drug uptake into this cell population, efficiency against the internal promastigotes and low systemic toxicity. Additional studies employed conjugated systems to increase certain properties of the product, for example AmpB-loaded chitosan-covered solid lipid nanoparticles.

### 4.2. Nanotechnology Applications

It is also worth highlighting that there have been continuing formulation developments using nanotechnology [22,25,27,28,46], such as nanoliposomes, nanocapsules, nanocochleates, cyclodextrins, and drug conjugates. This is particularly necessary for a molecule like AmpB which is poorly water soluble, poorly absorbed and toxic and not suitable for conventional pharmaceutical dosage forms. These formulations have shown unique advantages compared to the commercially available products such as reduced cytotoxicity, potential production cost reduction, enhanced targeted delivery, and reduction of the dose needed per treatment. It should be noted by changing the drug delivery system, that the mechanism of action may also be altered which could potentially achieve a shorter treatment duration, reduced drug doses and toxic effects, improved adherence, reduced production costs and delay emergence of resistant strains. These complex dosage forms meet specific needs related to maximizing the safety and efficacy of AmpB, but may be constrained by development costs and market forces. In addition, there are many others in development whose properties and benefits are still under study. Most formulations have been studied at early stages as a proof of concept using the intravenous route of administration, some of which are described in Table 3. The formulations in development use mainly two different approaches. The first approach is using a polymeric porous matrix to allow AmpB adsorption. Materials such as PEG, PBC, PUC, chitosan and PLGA, are some of the polymers that are being used to design new IV AmpB formulations. Other research teams use lipids for increased accessibility through cell membranes, stability, reduced production costs among other attractive features. Formulation components shown in Table 3 include Tween 80, phophatidylcholine, cholesterol, miglyol, and medium chain glycerides to achieve solubilization and pharmacokinetic modulation. The polymer formulation approach generally achieves a decrease of the AmpB-associated hemolytic activity, and a noticeable reduction of the dose needed per treatment; however, some of these formulations have a reduced efficiency in comparison with commercially available products, presumably due to a lack of drug release in-situ. As for the lipidic formulation approach, a significant toxicity reduction has also been achieved; nevertheless, compared with the polymeric approach, the selectivity and cellular uptake of the lipidic drug delivery systems are considerably higher. The high selectivity and potential dosage reduction for treatment are promising features.

Furthermore, nanotechnology development has enabled researchers to design ever more novel formulation designs. With this technology it is possible to design a material’s matrix to enable/enhance drug entrapment and allow specific targeting with surface functionalization or to modify the administration route as well. Oral drug administration is a very attractive option as well due to its higher patient adherence, relatively lower costs compared to parenteral dosage forms, minimum invasiveness, ease of use, and reduction of all the specialized associated expenses (health professionals, cold chain shipping and storage, hospital materials). Therefore, many researchers are focusing on developing an effective oral formulation. Furthermore, the use of lipids has been of particular interest in the oral delivery field because of their properties to improve drug solubility, mucosa penetration, lymphatic transport, and hepatic metabolism [47]. Applying the novel strategies to AmpB has resulted in new oral formulations for AmpB to treat leishmaniasis and candidiasis [29,30,31,33,35,36,37,38,39,40,41,48]. Studies so far report an improved absorption, increased drug uptake, reduced toxicity, as well as efficiency challenges due to poor drug release in some cases.

## 5. Oral AmpB Formulations

To date there are several oral AmpB formulations at various stages of preclinical development (Table 5) but few have made it to human clinical investigations [43,44,49,50,51,52,53,54,55,56]. Significant efforts have been made to develop lipid-based delivery systems for AmpB, such as solid lipid nanoparticles and self-emulsifying systems, to overcome solubility problems and enhance bioavailability. Polymeric nanoparticle and prodrug approaches will also be discussed.

### 5.1. Self-Emulsifying Drug Delivery Systems (SEDDS)

The oral AmpB lipid-based formulation developed by Wasan et al. was designed to reduce the limitations associated with existing intravenous formulations in treating systemic fungal infections [24,57,58,59,60,61,62,63] and VL in the developing world [17,18,64]. The rates of opportunistic fungal infections such as candidiasis, histoplasmosis and aspergillosis are climbing, particularly for immunosuppressed patients such as those with cancer, diabetes, or HIV/AIDS, or who are organ transplant recipients. Furthermore, the ability to self-administer the drug for either treatment or maintenance of treatment for systemic fungal infections would significantly increase the utility of this drug. It would also increase accessibility of this treatment in many geographic areas where refrigeration may not be available. This formulation, known as iCo-010, is based on a self-emulsifying mixture of monoglyceryl oleate (Peceol^®^, Gattefossé Canada, Saint-Laurent, QC), comprised of mono-, di and triglycerides of oleic acid (C18:1) acid, with lauroyl polyoxyl-32 glyceride (Gelucire^®^ 44/14, Gattefossè) and D-α-tocopheryl polyethylene glycol succinate (vitamin E-TGPS). The latter is an emulsifier and may serve as a penetration enhancer. The formulation also stabilizes the less-toxic monomeric form of AmpB (E. Wasan, unpublished results). In vitro and in vivo efficacy against VL [64] and candidiasis [65,66] demonstrated high activity comparable to liposomal AmpB [64,67]. This affect is related more to tissue accumulation of AmpB upon multiple dosing rather than a high C_max_ [68]. Notably, tissue levels in the kidney were relatively low in mice administered oral AmpB iCo-010, indicating that altered biodistribution plays an important role in the reduced toxicity observed with iCo-010. Safety and tolerability of this oral Amp B formulation following single and multiple-dose oral administration was demonstrated in healthy beagle dogs in GMP toxicology studies [69], representing an important first step toward its clinical development. Initial data from both cell line and in vivo research indicate that it is highly efficacious and exhibits low toxicity within the dosage range required for the treatment of diseases such as disseminated fungal infections and leishmaniasis [51,64,68,70].

Two human phase I clinical studies have been recently completed. In the phase 1a human clinical study, the primary endpoint of safety and tolerability of our oral AmpB formulation in capsule form (iCo-019) following administration of all single ascending doses were met including no signs of kidney, liver and gastrointestinal (GI) toxicities of note [71]. In addition, the iCo-019 oral AmpB formulation achieved a median plasma C_max_ of 28 ng AmpB/mL and AUC_(0-inf)_ of 1030 h·ng/mL at the lowest dose of 100 mg. At the 400 mg dose a median AUC_(0-inf)_ of 2029 h·ng/mL was achieved representing an approximate doubling of the AUC measure at an increased dose [71].

In the phase 1b human clinical study all repeated doses of our oral AmpB formulation were well tolerated with no serious adverse events including no signs of GI, kidney, and liver toxicities. Our oral AmpB formulation at the 100 mg dose achieved a median plasma C_max_ of 26 ng AmpB/mL and AUC_(0-inf)_ 991 h·ng/mL after day 1 of dosing and a median plasma C_max_ of 44 ng AmpB/mL and AUC_(0-inf)_ 1998 h·ng/mL after 10 days of dosing. This approximate doubling of the AUC_(0-inf)_ measure between day 1 and day 10 was observed not only at the 100 mg dose but at the 400 mg dose as well [72].

### 5.2. Self-Nanoemulsifying Drug Delivery Systems (SNEDDS)

The use of SNEDDS for the enhancement oral bioavailability is well established, with various Food and Drug Administration (FDA) approved formulations such as Norvir^®^ (ritonavir), Sandimmune^®^ (cyclosporine) and Rapamune^®^ (cyclosporine) [38]. SNEDDS have a submicron emulsion droplet size, whereas SEDDS may have larger droplets. The smaller droplet size may enhance oral absorption by facilitating micellization in the intestine.

SNEDDS are highly versatile and are amenable to the incorporation of various functional excipients as demonstrated in a recent study by Kontogiannidou et al. [73] whereby SNEDDS were co-formulated with the polymer, tri-methyl chitosan for the enhancement of mucoadhesion and permeation. The authors encapsulated AmpB within SNEDDS formulated using Captex^®^ 355 (ABITEC, Columbus, OH, USA), Kolliphor^®^ RH40 (Sigma-Aldrich, Saint Louis, MO, USA) and propylene glycol as the oil phase (45%), emulsifier (45%) and co-solvent (10%) respectively. In an in vitro model using Caco-2 monolayers, the addition of tri-methyl chitosan (0.25–0.5%) was associated with modest permeability increments of up to 11% in comparison to plain AmpB loaded SNEDDS [74]. As the geographical spread of leishmaniasis is largely confined to tropical countries with constrained resources, the stability of AmpB formulations under room temperature conditions is a critical product parameter and it was observed that the incorporation of tri-methyl chitosan enhances the stability of AmpB, compared to plain SNEDDS under room temperature conditions [74].

In a separate study, the same authors utilized the same components (Captex^®^ 355, Kolliphor^®^ RH40, 1;1 (*w*/*w*) with 10% propylene glycol as co-surfactant) to investigate the utility of incorporating room temperature ionic liquids (RTILs) in SNEDDS to further enhance the solubility and permeability of AmpB [74]. RTILs are hydrotropic organic salts and can improve the solubility of hydrophobic drugs. A series of RTILs based on 1-butyl-3-methylimidazolium tetrafluoroborate, cationic imidazolinium salts, was used. The AmpB-RTIL in SNEDDS (3:7 *v*/*v*) was prepared by mixing to homogeneity to form the pre-concentrate SNEDDS. In this study, the authors investigated the effect of alkyl chain length and the type of ion of imadizolium based RTILs on AmpB solubility and cytotoxicity [74]. AmpB has an aqueous solubility of less than 0.001 mg/mL [75], and in both studies, by utilizing SNEDDS, AmpB solubility was enhanced to 0.67 mg/mL, with the addition of RTILs producing more a more pronounced solubility enhancement of up to 1.67 mg/mL in simulated gastric fluid [76]. From the study, the authors concluded that increasing the alkyl length and type of anion (PF6- > BF4- > Cl-) in RTILs was positively correlated with AmpB solubility and permeability enhancements in vitro [74]. The authors also observed that under simulated gastric conditions or ambient conditions, the addition of RTILs to SNEDDs significantly enhanced AmpB stability more than plain SNEDDS [74]. AmpB in RTIL was demonstrated to cross Caco-2 cell monolayers but some toxicity was observed in vitro for the SNEDDS formulation. Combined with the low solubility relative to that needed for clinical studies, the authors suggest further development is forthcoming, including in vivo characterization. A general representive diagram depicting the formulation approaches and the mechanism of drug absorption of orally administered AmpB-SNEDDS formulations is presented in Figure 3.

### 5.3. Cochleate Formulations

Cochleates are stable lipid nanoparticles that are characterized by the presence of multiple spiral-shaped lipid bilayers consisting of negatively charged phospholipids such as phosphatidylserine, phosphatidylcholine, phosphatidic acid, phosphatidylglycerol or phosphatidylinositol [77,78]. The bilayers are tightly packed together through electrostatic interactions due to the addition of divalent cations such as Mg^2+^ or Ca^2+^ which act as bridging agents and can result in a dehydrated interior environment in contrast to liposomes which have an aqueous core [78,79]. Hydrophobic drugs such as AmpB may be entrapped within the phospholipid bilayers, which shields the entrapped drug from degradation in harsh environments such as bile salt or acidic conditions. The cochleate structure is dismantled after cellular uptake due to differential cation gradients between the cochleate and the cell which ultimately results in intracellular drug release as well [80]. Due to their hydrophobic core nature and considerable stability, cochleates have been one the most investigated oral drug delivery vehicles for AmpB [81,82,83,84].

The synthetic version of phosphatidylserine (1,2-dioleoyl-sn-glycero-3-phospho-L-serine (DOPS) has been widely used in cochleate formulation studies [85], but it is relatively expensive which may limit accessibility if incorporated in commercial formulations [86,87]. In a study by Lipa-Castro et al. [51], the authors formulated AmpB-loaded cochleates based on DOPS, cholesterol and Vitamin E as an antioxidant, and performed physical characterization of the cochleate structure. This included confirmation of the less-toxic monomeric form of AmpB as well as stability in gastric fluid. In a follow-up study [52], a murine model of VL was utilized and it was observed that intraperitoneally administered AmBisome^®^ (Gilead Sciences, Inc., Foster City, CA, USA), empty cochleates and orally administered AmpB-loaded cochleates (1 mg/kg/48 h × 3 doses) reduced liver parasitic load by 77%, 24% and 37% respectively, with the limited efficacy of the oral formulation highlighting the need for further optimization to match the current standard in antileishmanial therapy.

To optimize the formulation and alleviate the cost challenges associated with the use of DOPS, the authors sought to utilize soy derived phosphatidylserine (Lipoid^®^ PSP70, UL Prospector, Northbrook, IL, USA) instead, to formulate orally administered AmpB-loaded cochleates [86,87]. By utilizing Lipoid^®^ PSP70, AmpB was still entrapped within the cochleates, with reported entrapment efficiencies of up to 65% and of note, spectral analysis revealed that the developed formulation was characterized by different aggregation behaviour to Fungizone^®^ [86]. Due to the need to circumvent the adverse nephrotoxic effects of Fungizone^®^, there is a need to formulate AmpB in an aggregation state that is distinctly different from that of Fungizone^®^ or in monomeric form.

An in vitro study utilizing Caco2 cells demonstrated that the nano-cochleate formulations had lower toxicity than the Fungizone^®^ formulation demonstrating the utility of the cochleate formulation [87]. In subsequent in-vivo studies, the authors observed that loading of Lipoid^®^ PSP70 cochleates in enteric coated capsules generated higher anti-leishmanial activity in mice compared to cochleate administration as a suspension [87]. The enhanced activity with the enteric coated capsules was attributed to better protection against AmpB degradation from the acidic conditions in the stomach [87]. Although the authors were able to generate comparable antileishmanial activity with a less expensive phospholipid than DOPS and were able to formulate cochleates with an acceptable safety profile, the antileishmanial activity of this formulation is still limited.

Cochleate formulations do hold great potential as a safe and efficacious oral drug delivery method for AmpB. Matinas^®^ Biopharma (Bedminster, NJ, USA) recently published results demonstrating the safety and tolerability of an orally administered formulation of a proprietary encochleated AmpB deoxycholate formulation (MAT 2203) (1–2 g dose) in HIV patients with a history of cryptococcal meningitis [80]. Currently, the company is undertaking Phase II trials (Clinical trials.gov registration no. NCT04031833) to determine the efficacy of the oral formulation for cryptococcal meningitis, with an estimated study completion date of October 2022. A summary of the company’s efforts in utilizing MAT 2203 for other fungal diseases such as systemic candidiasis and invasive aspergillosis have recently been published by Aigner and Lass-Florl [88].

### 5.4. Solid Lipid Nanoparticles

Solid lipid nanoparticles (SLNs) are lipid-based nanocarriers that utilize lipids that are solid at room temperature and their use is associated with advantages such as solvent free-formulation methods, higher drug stability and loading efficiency in contrast to other lipid based carriers such as liposomes [89]. The encapsulation of AmpB within SLNs has been demonstrated to be a formulation approach that can enhance AmpB oral bioavailability and largely preserve AmpB in the more ergosterol-selective and less toxic monomeric state in comparison to the dimeric state found in the commercial formulation, Fungizone^®^ [90,91].

The conjugation of vitamin B_12_ to nanoparticles has been demonstrated to enhance the oral bioavailability of different drugs, including insulin [92,93], and Singh et al. [94] recently utilized this formulation approach in an aim to further enhance the oral bioavailability of AmpB loaded SLNs. By coating a vitamin B12-stearic acid conjugate onto AmpB encapsulated SLNs based on the excipients; glycerol monosteate, Precirol^®^ ATO (Gattefossé, Saint-Priest, France), Pluronic^®^ F127 (Sigma-Aldrich, Saint Louis, MO, USA) and Solutol^®^ (Sigma-Aldrich, Saint Louis, MO, USA), the authors developed a formulation with superior in-vitro cell viability, cellular uptake and anti-leishmanial activity compared to AmpB alone [94]. For the in-vitro evaluation of anti-leishmanial activity, *L. donovani*-infected J774A.1 macrophages were utilized and the vitamin-B12-stearic acid coated SLNs achieved IC_50_ values that were 3-fold better than AmpB alone [94].

In a parallel study, the same authors also explored the utility of co-formulating of AmpB with another anti-parasitic agent (paromomycin) for the generation of synergistic anti-leishmanial activity [95]. Using an emulsion/solvent evaporation method, the authors formulated 2-hydroxypropyl-β-cyclodextrin (2-HPCD) modified SLNs based on the excipients; glycerol monostearate and soybean lecithin [95]. In an in-vitro model of anti-leishmanial activity with *L. donovani*-infected macrophages, the 2-HPCD-AmpB SLNs achieved superior IC_50_ values that were 23-fold and 14-fold lower than free AmpB and AmBisome^®^ alone [95]. The utility of the co-formulation approach using HPCD-modified SLNs was further demonstrated in another study whereby the authors combined AmpB with melatonin for synergistic activity [96]. Similar to the other study, superior anti-leishmanial activity was observed in vitro with the co-formulation of 2-HPCD AmpB/melatonin, with IC_50_ values 62-fold better than free AmpB [96].

Additionally, in both studies, there were no significant elevations observed with respect to nephrotoxicity and hepatoxicity markers in vivo demonstrating the potential of the formulation as a safe oral formulation [96]. Furthermore, in both studies, the HPCD-modified SLN formulations demonstrated significantly higher anti-leishmanial activity by reducing the liver parasite burden compared to negative controls, oral miltefosine as a positive control, and non-modified SLN formulations containing AmpB [95,97]. For instance, in one of studies, in an in vivo model of *L. donovani*-infected mice, the co-formulation of AmpB and melatonin in SLNs inhibited liver parasite burden by up to 86% in contrast to melatonin-SLNs and AmpB-SLNs with 40% and 59% inhibition respectively [96]. Of note, the addition of 2-HPCD to the AmpB-Melatonin-SLNs enhanced the parasite inhibition up to 98% following oral dosing at 10 mg/kg day × 5, which was significantly higher efficacy than the positive control of oral miltefosine [96].

The inclusion of cyclodextrin enhanced epithelial permeation and enabled absorption through micropinocytosis and the caveolae-mediated route of endocytosis [97]. In spite of the promising results with respect to the formulation’s efficacy, significant changes in the particle sizes of the HPCD formulations under storage conditions of 40 °C and 25 °C were observed, which may warrant further stability investigations to elucidate any changes on AmpB aggregation state under storage and the ultimate effects on efficacy and safety [95].

### 5.5. Polymer-Based Formulations

In a study by Kaur et al. [53], the authors investigated the utility of polymeric ethylcellulose-based nanoparticles in enhancing the oral bioavailability of AmpB. Using a high-pressure emulsification solvent evaporation method to encapsulate AmpB, the authors achieved an encapsulation efficiency of up to 60%, enhanced bioavailability of AmpB 11-fold in Witsar rats and preserved AmpB in a monomeric state that was distinctly different from Fungizone^®^ [53]. In the study, the encapsulation of AmpB within ethylcellulose nanoparticles was associated with lower haemolytic toxicity in-vitro and lower levels of nephrotoxicity markers in-vivo compared to pure AmpB and Fungizone^®^ [53]. Additionally, the ethylcellulose formulation was associated with greater antifungal efficacy (*Candida albicans*) in comparison to pure AmpB and Fungizone^®^ [53]. These results taken in conjunction with the demonstrated oral uptake in rats and in vivo safety indicate the potential of ethylcellulose based formulations for oral delivery of AmpB, though additional studies are required to determine in-vivo antifungal efficacy.

### 5.6. Pro-Drug Approach

One of the greatest pitfalls associated with current AmpB treatment is the incidence of adverse events such as nephrotoxicity and hepatoxicity, which have been directly correlated to the aggregation state of AmpB [18]. By conjugating an amine version (olelyl-amine) of the long chain fatty acid, oleic acid, to AmpB, Thanki et al. [50], synthesized a pro-drug version of AmpB with different aggregation behaviour to pure AmpB. The oleyl-amine AmpB pro-drug by itself did not result in the induction of hepatotoxic or nephrotoxic effects as demonstrated by in-vitro and in-vivo studies in contrast to pure AmpB and Fungizone^®^ [50]. In addition, AmpB prodrug was associated with higher in vitro antifungal efficacy and higher oral bioavailability with pharmacokinetic studies in mice demonstrating 3 and 4-fold increments in C_max_ and AUC in comparison to pure AmpB [98,99].

In another study, of the oleyl-amine AmpB prodrug within a SNEDDS formulation based on the excipients, Capmul^®^ MCM C8 (UL Prospector, Northbrook, IL, USA), Cremophor^®^ RH 40 (UL Prospector, Northbrook, IL, USA) and propylene glycol as the oil phase, surfactant and co-surfactant respectively further enhanced the C_max_ and AUC of the prodrug by 70% and 80% respectively [98,99]. Of note, the non-toxicity of the pro-drug was maintained in vivo for the SNEDDS formulation demonstrating the potential clinical applicability of the developed formulation [99]. Additionally, accelerated stress testing at 40 °C/75% relative humidity revealed that the formulation was sufficiently robust with respect to drug content and particle size [99]. The stability data of the pure pro-drug and SNEDDS formulation coupled with the demonstrated absorption enhancement and safety profile in-vivo and in vitro are promising, however in vivo studies still need to be conducted to determine the efficacy of this pro-drug approach [50,98,99].

## 6. Veterinary Applications of Amphotericin B

AmpB is also used in a wide variety of systemic fungal infections companion animals including aspergillus [100], coccidioidomycosis [101], cryptococcus [102], sporotrichosis [103], leishmaniasis [104], and blastomycosis [105]. Newer antifungals are expensive thus impractical for veterinary use, therefore amphotericin B remains an appropriate treatment option [106]. It is typically reserved for severe, life-threatening infections [101,106,107] or infections refractory to azole treatment [103,107] due to toxicity risk. A major drawback to use is nephrotoxicity, especially in the original sodium deoxycholate dispersion [107]. Monitoring of renal function is necessary because the development of renal dysfunction while receiving AmpB is common in cats and dogs, and often warrants temporary or complete discontinuation [101,108,109]. Dose is based on type of infectious organism and appropriate maximal cumulative dose, animal type and weight; current AmpB formulations may be given either intravenously or subcutaneously [107,109]. Oral AmpB has been used in the treatment of *Macrorhabdus ornithogaster*, an infection of the gastrointestinal tract in birds [110,111] by dissolving AmpB in water or lactulose for oral administration for local treatment [110,111].

### Amphotericin B in Treatment of Canine Leishmaniasis

Treatment of canine Leishmaniasis (CanL) has been the subject of debate. The primary goal of veterinary treatment is to ultimately reduce disease burden in human populations, and complete eradication of infected animals remains a questionably necessary option when considering the gravity of human infection. In many endemic countries, domestic dogs are the most important reservoir host for the Leishmania protozoa. Infection may spread to humans via bite of infected sandflies [22]. There has been increased reports of resistance in leishmaniasis to AmpB in the past decade [112,113], which poses increased risk on human patients as leishmaniasis is fatal if untreated [22]. Several treatment guidelines for CanL warn of inevitable relapse following treatment with AmpB suggesting overall low efficacy in disease eradication [109,114]. The WHO Expert Committee on the Control of Leishmaniases does not recommend use of AmpB to treat CanL, a large number of dogs remain infectious despite semblance of clinical healing allowing for continued use as reservoir host by the pathogen; and with overall low efficacy, increased use promotes resistance [115].

Several studies examining efficacy of AmpB in treatment of CanL saw resolution of skin lesions and reduction in lymphadenomegaly 2 months post treatment [104,116]. In one study, Olivia et al. [104], 13 dogs naturally infected with CanL were treated with AmBisome^®^ total cumulative dose of 10–15 mg/kg given over 3–5 infusions over 10 days. Several dogs experienced temporary increase in serum creatinine and urea, which normalized rapidly post treatment. Despite early apparent clinical success, 12 of the 13 dogs in this study tested positive for leishmania in lymph node aspirate at 2 months post treatment and relapsed at 4–6 months [104]. In a similar study, 19 dogs with CanL were treated with amphotericin B deoxycholate combined with Intralipid, total cumulative dose of 10–17.7 mg/kg over 10 infusions given twice weekly [116]. During treatment, two dogs died from unrelated infection and unknown causes, and one dog had to stop treatment for 20 days due to rise in serum creatinine. After completion of AmpB treatment, all dogs were given oral allopurinol 20 mg/kg/day for 7 days a month to prevent relapse [117]. In this study, 14 of the 17 dogs were PCR negative for leishmaniasis in bone marrow 3 months post treatment.

In another study, 16 dogs were treated with CanL with Amp-B deoxycholate diluted with water and soybean oil given at doses of 0.8–2.5 mg/kg via twice weekly infusions for 8–10 sessions [118]. Several dogs experienced rise in serum creatinine which temporarily postponed treatment for 10 days, this treatment delay is common in many AmpB CanL intervention studies [104,116]. After treatment was complete, all remaining dogs were negative for leishmaniasis in bone marrow. Four dogs remained PCR negative at 18 months, five were PCR negative as well but lost to follow up before 1 year; many remaining dogs experienced relapse and were treated with allopurinol, left the study, or were euthanized.

From these studies, it can be observed that while AmpB provides rapid clinical improvement, maintenance of clinical cure remains elusive, and long-term follow up is necessary to gauge overall treatment success. Differences in patient characteristics, severity of initial infection, treatment protocol, or use of allopurinol [117] as secondary prevention following treatment with AmpB may impact the successful treatment of canine leishmaniasis. The availability of oral AmpB would improve convenience and access to treatment for CanL, allowing for administration by untrained animal caretakers; however, further research into effective combined-therapy treatment regimens and secondary protocols for maintenance of remission, such as oral secondary prophylaxis, remain necessary. Appropriate guidelines for defining remission, treatment failure, and monitoring parameters should be established to follow good antimicrobial stewardship in light of the WHO recommendations against use of AmpB for CanL and concerns of growing resistance.

## 7. Future Perspectives and Conclusions

As we discussed in our recent reviews [17,18], AmpB is a polyene macrolide antibiotic administered intravenously in the treatment of a variety of systemic fungal infections including candidiasis, aspergillosis, fusariosis, and zygomycosis [57]. In addition, AmpB has exhibited antiparasitic activity for certain protozoan infections, including leishmaniasis as well as primary amoebic meningoencephalitis [58]. Prior to the development of lipid-based formulations, the commercially available formulation used in the clinic was Fungizone^®^, a conventional micellar form of AmpB in a complex with deoxycholate [59]. Alternative or lipid-based formulations have been developed to overcome some of the toxicity problems associated with the conventional formulation. There are several lipid-based parenteral formulations which have been marketed to treat fungal infections, which include the liposomal formulation AmBisome^®^, the lipid complex formulation Abelcet^®^, and a colloidal dispersion formulation Amphocil^®^ (Amphotec) [61,62,63]. More recently, an emulsion form of AmpB (Amphomul^®^) was developed and completed its Phase III clinical trial in 2014 [24]. The aim of this trial was to assess the safety and efficacy of the parenteral lipid emulsion formulation compared to AmBisome^®^ as a single infusion treatment for VL [24]. However, its use has been limited by dose-dependent nephrotoxicity and parenteral administration [17,18,19,20] which may be inaccessible to many, expensive, difficult to administer to patients requiring appropriate medical personal and sterile conditions, and formulation stability in non-refrigerated conditions.

To overcome these challenges the development of an oral formulation of AmpB that is cost-effective, easy to administer, non-toxic yet retaining pharmacological activity and the ability to store at room temperature would be ideal [17,18]. However, to date few oral formulations of AmpB have been developed [43,44,49,54,56] but interest remains high due to the continuing need for efficacious therapy that are accessible.

## Figures and Tables

**Figure 1 pharmaceutics-14-02316-f001:**
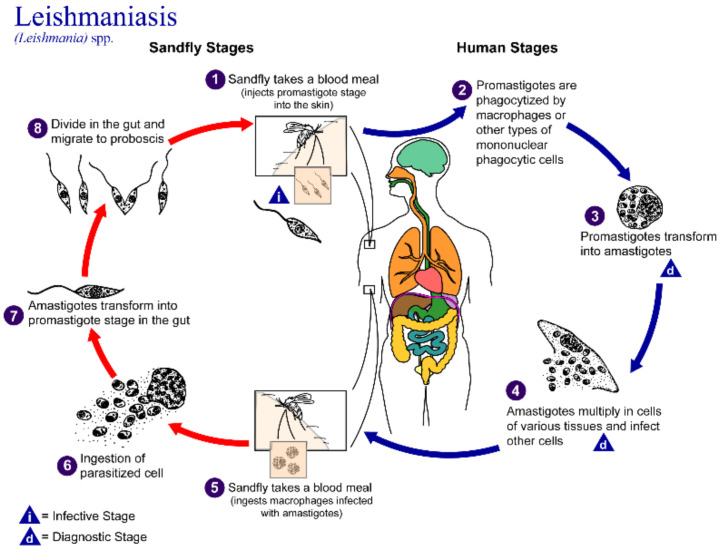
Life cycle of the Leishmania parasite. Reprinted with permission from https://www.cdc.gov/parasites/leishmaniasis/biology.html. CDC/Alexander J. da Silva, Ph.D. (accessed on 17 October 2022) [1].

**Figure 2 pharmaceutics-14-02316-f002:**
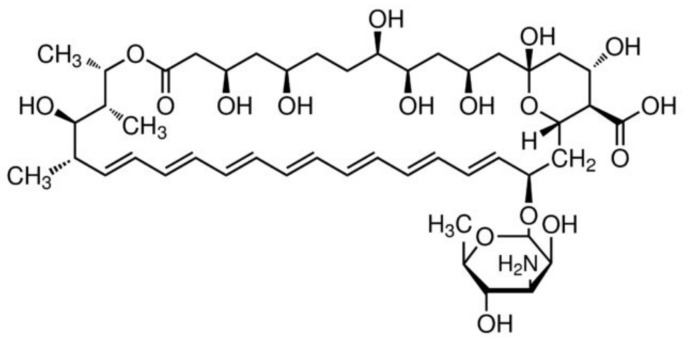
Chemical structure of Amphotericin B.

**Figure 3 pharmaceutics-14-02316-f003:**
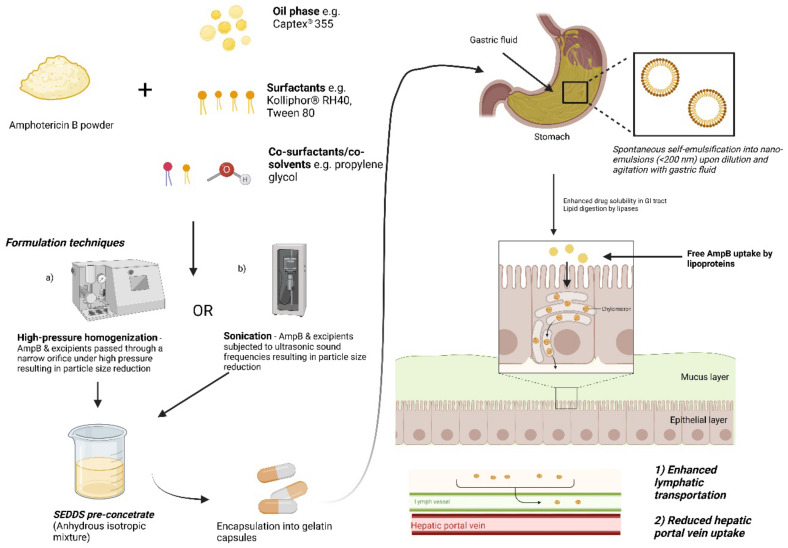
Schematic depicting the general formulation approaches and solubility enhancement of AmpB loaded in self-emulsifying systems. Adapted from [76]. Image generated and adapted from “Lipid handling in the small intestine modulates immune system homeostasis” by biorender.com. Retrieved (2020). Retrieved from https://app.biorender.com/bio-render-templates (accessed on 17 October 2022).

**Table 1 pharmaceutics-14-02316-t001:** Uses of Amphotericin B (systemic) Source: Adapted from DrugBank Online [9].

Coccidioidomycosis	Ocular aspergillosis
Fungal infections	Refractory aspergillosis
Histoplasmosis	Severe Coccidioidomycosis
Invasive Aspergillosis	Severe Cryptococcosis
Invasive Fungal Infections	Severe Fungal infection: *Basidiobolus* spp.
Leishmaniasis	Severe Fungal infection: *Conidiobolous* spp.
Meningitis, Cryptococcal	Severe Fungal infection: *Sporotrichosis* spp.
Meningitis, Fungal	Severe Histoplasmosis
Mucocutaneous Leishmaniasis	Severe Mucocutaneous leishmaniasis
Mycotic endophthalmitis	Severe North American blastomycosis
*Penicillium marneffei* infection	Severe Systemic candidiasis
Visceral leishmaniasis	Ocular aspergillosis
Candidal cystitis	Refractory aspergillosis
Disseminated Cryptoccosis	Severe Coccidiomycosis
Fungal osteoarticular infections	Severe Cryptococcosis

**Table 2 pharmaceutics-14-02316-t002:** Commercial parenteral formulations of AmpB.

Formulation	Excipients	PharmacokineticFeatures:t_½_, t_1/2β_, V_d_	Dosage (mg/kg)	References
D-AmpB	NaDC	24 h/ 15 days/ not reported	0.7–1 mg/kg	Thakur et al., 1999 [11]
Fungizone^®^				
Anforicin^®^				
ABLC (Abelcet^®^)	DMPC DMPG	24 h/ 10 days/ 131 ± 58 L	5 mg/kg	Stevens 1994 [12]
ABCD (Amphotec^®^)	Disc-shaped AmpB cholesteryl sulfate complex	24 h/ 4–8 weeks/ not reported	2 mg/kg	Guo 2001 [13]
L-AmpB	HSPC Cholesterol DSPG α-tocopherol	24 h/ 6 days/ 18.9–49.1 L	3 mg/kg	Adler-Moore and Proffitt 2002 [14]; Bern et al., 2006 [15]
Liposomal AmpB				
AmBisome^®^				

**Table 3 pharmaceutics-14-02316-t003:** Parenteral formulations of AmpB in preclinical development.

Formulation	Excipients	Advantages	Targeted Species	References
AmpB/B and AmpB/U	PEG-PBC and PEG-PUC	Less hemolytic activity	Not indicated	Wang et al., 2016 [29]
DSHemsPC-AmpB-Lip	1,2-Distigmasterylhemisuccinoyl-sn-glycero-3 phosphocholine DMPC DMPG	Reduced production cost and nephrotoxicity. In vivo studies in a mouse model	*L. major*	Iman et al., 2017 [30]; Iman et al., 2011 [31]
PLGA-AmpB	PLGA	Dosage reduction and increased efficacy in a mouse models.	*L. infantum*	Van de Ven et al., 2012 [32]
MPPIA-AmpB	Poly(propylene imine)dendrimer conjugated with mannose	Reduced toxicity. Macrophage targeting. Increased cellular uptake.	*L. donovani*	Jain et al., 2015 [33]
LcPGNP-AmpB	Glycoprotein Lactoferrin PLGA	Nanoparticle formulation	*L. donovani*	Asthana et al., 2015 [34]
CHOL-NE-AmpB	Medium chain triglycerides Tween80 Cholesterol α-tocopherol	Increased selectivity and stability.	*L. amazonesis/L. infantum*	Caldeira et al., 2015 [35]; Santos et al., 2018 [36]
ME-AmpB	Mygliol^®^ 812 Tween80 Lipoid^®^ S100	Increased efficacy and selectivity in a mouse model	*L. donovani*	Rochelle et al., 2018 [37]
NQC-AmpB	Chitosan chondroitin sulfate	in vivo efficacy in a mouse model.	*L. amazonesis*	Ribeiro, Chavez et al., 2014 [38]; Ribeiro, Franca et al., 2014 [39]
AmpB-C-SLNs	Chitosan stearic acid soy-phosphatidylcholineTween80	Reduced toxicity and increased efficiency in vivo in a mouse model	*L. donovani*	Jain et al., 2014 [40]
PLGA-PhoS-AmpB	PLGA decorated with 3-O-sn-Phosphatidyl-L-serine	Increased stability and efficacy in vivo in a mouse model	*L. donovani*	Singh et al., 2018. [41]

**Table 4 pharmaceutics-14-02316-t004:** Preclinical formulations of AmpB with macrophage targeting moeities.

Formulation	Composition	Administration Route	Targeted Species	References
MTC-AmpB	Mannose-anchored thiolated chitosan NPs for AmpB and Tween80	oral	*L. Donovani*	Sarwar et al., 2018 [43,44]; Shahnaz et al., 2017
MPPIA-AmpB	Poly(propylene imine)dendrimer conjugated with mannose + AmpB	Intravenous	*L. Donovani*	Jain et al., 2015 [45]
MnosCNc-AmpB	AmpB entrapped mannose grafted chitosan nanocapsules	Intravenous	*L. Donovani*	Asthana et al., 2015 [46]
AmpB-PM	Amphotericin B and Polymannose conjugate.	Intravenous	*C. Albicans*	Francis et al., 2018 [47]
MTC AmpB	mannose-anchored thiolated chitosan AmpB nanocarrier	Intravenous	*L. donovani*	Shahnaz et al., 2017 [48]

**Table 5 pharmaceutics-14-02316-t005:** Novel oral AmpB Formulations.

Formulation	Excipients	Targeted Species	References
MTC-AmpB	Mannose-anchored thiolated chitosan NPs Tween80	*L. donovani*	Sarwar et al., 2018 [43]; Shahnaz et al., 2017 [44]
CopNEC-AmpB	D-α-tocopherol polyethylene glycol 1000 succinate phosphatidylcholine Copaiba oil	*L. donovani*	Gupta et al., 2015 [49]
AmpB-OA	oleic acid conjugated to AmpB	not established	Thanki et al., 2018 [50]
iCo-010	Peceol^®^ Gelucire^®^	*L. donovani*	Wasan et al., 2015 [51]
ChiAmp NLC	Chitosan Tween80 lecithin	not established	Ling et al., 2019 [52]
AmpB-EC-NPs	Ethyl cellulose	*C. albicans*	Kaur et al., 2020 [53]
Chitosan coated PLGA containing AmpB	Chitosan-coated PLGA	*C. albicans* *C. tropicalis* *C. glabrata*	Ludwig et al., 2018 [54]
Trag-AAc-AmpB	Tragacanth Acrylic Acid	*C. albicans*	Mohamed et al., 2018 [55]
CAMB	Phosphatidylserine Calcium	*C. albicans*	Desai et al., 2022 [56]

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
