# Peer review of "Review of Novel Oral Amphotericin B Formulations for the Treatment of Parasitic Infections"

_pharmaceutics, 2022, doi:10.3390/pharmaceutics14112316_

Round 1

Reviewer 1 Report

Authors have written a review article titled “Review of Novel Oral Amphotericin B Formulations for the Treatment of Parasitic infections”. The manuscript lacks some important information. The manuscript can be considered for publication after major revision with consideration following points.

1.      In the introduction section, the current status of the global parasitic infection is missing.

Additionally, please add challenges faced by presently available therapeutics for treating parasitic infection.

2.      The authors have mainly discussed the use of nanoparticulate formulation /novel drug delivery system for the delivery of Amphotericin B. Apart from these formulations, there are many conventional formulations are available in the market containing Amphotericin B. How these nanoparticulate formulations/novel drug delivery systems are superior to the conventional formulation ?,

3.      A table summarizing all the developed oral Amphotericin B formulations, their composition and clinical uses must be included to enhance the quality of the manuscript.

4.      The manuscript does not contain any graphical/schematic representation of the novel oral Amphotericin B formulations, preparation methods, and/or mechanism of drug delivery or action. If authors are able to add above mentioned diagrams/schematics, it will be highly appreciable

Author Response

Reviewer #1: 
1. In the introduction section, the current status of the global parasitic infection is missing. Additionally, please add challenges faced by presently available therapeutics for treating parasitic infection.
Response: In the revised manuscript we have included a section on the status of global parasitic infections (specifically leishmaniasis since amphotericin B is used primarily for this disease) and the current challenges faced by presently available therapeutics for treating parasitic infections.  

2. The authors have mainly discussed the use of nanoparticulate formulation /novel drug delivery system for the delivery of Amphotericin B. Apart from these formulations, there are many conventional formulations are available in the market containing Amphotericin B. How these nanoparticulate formulations/novel drug delivery systems are superior to the conventional formulation?
Response: There several recent review articles that discuss how these nanoparticle formulations/novel drug delivery systems are superior to current conventional formulations. We have included these references in our revised manuscript. Many of these formulations are cost-effective, safe, tropically stable and accessible compared to the current parenteral formulations. 

3. A table summarizing all the developed oral Amphotericin B formulations, them composition and clinical uses must be included to enhance the quality of the manuscript.
Response: Table 2 of our manuscript does summarize all the developed oral amphotericin B formulations. In addition, several recent review papers by our group have been published that highlights the composition and clinical uses of these oral amphotericin B formulations. We have included these references in our revised manuscript: 

Cuddihy G, Wasan EK, Di Y, Wasan KM. The Development of Oral Amphotericin B to Treat Systemic Fungal and Parasitic Infections: Has the Myth Been Finally Realized? Pharmaceutics. 2019 Feb 26;11(3):99. doi: 10.3390/pharmaceutics11030099. PMID: 30813569; PMCID: PMC6470859.

Wasan KM. Development of an Oral Amphotericin B Formulation as an Alternative Approach to Parenteral Amphotericin B Administration in the Treatment of Blood-Borne Fungal Infections. Curr Pharm Des. 2020;26(14):1521-1523. doi: 10.2174/1381612826666200311130812. PMID: 32160842.

4. The manuscript does not contain any graphical/schematic representation of the novel oral Amphotericin B formulations, preparation methods, and/or mechanism of drug delivery or action. If authors are able to add above mentioned diagrams/schematics, it will be highly appreciable
Response: We have included a new schematic representation of the preparation of our novel oral amphotericin B formulation in the revised manuscript. In addition, we have recently published a paper that provides a graphical schematic representation of the novel oral amphotericin formulations, preparation methods and/or mechanisms of drug delivery. We have cited this review paper in this paper. This paper is focused on the role of these formulation in both animal and human parasitic infections: 

Cuddihy G, Wasan EK, Di Y, Wasan KM. The Development of Oral Amphotericin B to Treat Systemic Fungal and Parasitic Infections: Has the Myth Been Finally Realized? Pharmaceutics. 2019 Feb 26;11(3):99. doi: 10.3390/pharmaceutics11030099. PMID

Reviewer 2 Report

Ellen Wasan and co-workers reviewed and summarized the Oral Amphotericin B Formulations for the Treatment of Parasitic infections. The review contains several oral formulations and discusses the pharmacokinetics and formulation aspects. Furthermore, it sheds light on the need for such formulations. This work is interesting and adds to the existing knowledge in Amphotericin B Formulations.

The work is well organized overall and deserves to be published in the Pharmaceutics journal.

I appreciate the author's efforts in reporting. However, there are some errors that prevent it from acceptance in the current version.

Detailed comments

Major:

Please address these before the revision.

1.     Page 3 line 74-83. Reorganize the aims of the review and cite all the previous reviews on this topic and include the novelty of this review.

2.     Is the entire table 1 reference the same? Make this clear in legend.

3.     Lines 54-70 have no references. Cite each statement if it’s not your own.

4.     “Development of Oral Amphotericin B Formulations”

This section is interesting but very dry for the reader. Add figures from the literature you cited. At least first 5 headings.

For example, you can show how Pk has changed after formulation or how safety has improved. Visual impact is stronger than text. Authors may use their own published research also.

5.     SEDDS and SNEDDS discuss what is the difference in pK, globule size and composition.

6.     Veterinary Applications of Amphotericin B seems off-topic here. Confirm if all these were oral formulations only.

7.     Please include all existing formulations in a table and mention any ongoing clinical trials of the same.

8.     Why is it that all Amphotericin B formulations are complex technologies such as nanotech, and lipid tech, in this case, how do you think the price will be affordable? Are there any conventional simple formulations such as capsules or suspensions? Is it because of safety? Or because of the established status quo?

Minor:

1.     Use this acronym throughout the manuscript. Amphotericin B (AmpB). Do not use variations. Check all other abbreviations.

2.     Several species names. Use italics. Including tables.

3.     Use numbers for all your headings.

4.     Take care of subscripts for all PK terms, also IC50, etc.

5.     Include if the company had any role in this manuscript in conflicts statement. 

Author Response

Reviewer #2
Major:
Please address these before the revision.
1.    Page 3 line 74-83. Reorganize the aims of the review and cite all the previous reviews on this topic and include the novelty of this review.
Response: We have addressed this in the revised manuscript. One of the main novelties of this review paper is the focus on the treatment of not only human parasitic infections but also veterinary parasitic infections. 

2.    Is the entire table 1 reference the same? Make this clear in legend.
Response: We have addressed this in the revised manuscript. 

3.    Lines 54-70 have no references. Cite each statement if it’s not your own.
Response: We have addressed this in the revised manuscript. 

4. “Development of Oral Amphotericin B Formulations”
This section is interesting but very dry for the reader. Add figures from the
literature you cited. At least first 5 headings. For example, you can show how Pk has changed after formulation or how safety has improved. Visual impact is stronger than text. Authors may use their own published research also.
Response: The review brings up a very good point and we have tried to modify this section to make it more engaging for the reader. 

5. SEDDS and SNEDDS discuss what is the difference in pK, globule size
and composition.
Response: We have address this in the revised manuscript 

6. Veterinary Applications of Amphotericin B seems off-topic here. Confirm
if all these were oral formulations only.
Response: We have addressed this in the revised manuscript. One of the main novelties of this review paper is the focus on the treatment of not only human parasitic infections but also veterinary parasitic infections.

7. Please include all existing formulations in a table and mention any
ongoing clinical trials of the same.
Response: All existing oral amphotericin B formulations have been included in table 2 of the revised manuscript. 

8. Why is it that all Amphotericin B formulations are complex technologies
such as nanotech, and lipid tech, in this case, how do you think the price will
be affordable? Are there any conventional simple formulations such as
capsules or suspensions? Is it because of safety? Or because of the
established status quo?
Response: The Wasan lab formulation which has completed phase 1 clinical studies is in a simple capsule form. This formulation is cost-effective, safe, tropically stable and accessible with positive human phase 1 results and no adverse events reported. 

Minor:
1. Use this acronym throughout the manuscript. Amphotericin B (AmpB). Do not use variations. Check all other abbreviations.
Response: This has been corrected in the revised manuscript. 

2. Several species names. Use italics. Including tables.
Response: This have been corrected in the revised manuscript 

3. Use numbers for all your headings.
Response: This has been addressed in the revised manuscript. 

4. Take care of subscripts for all PK terms, also IC50, etc.
Response: This has been addressed in the revised manuscript. 

5. Include if the company had any role in this manuscript in conflicts statement. Response: The company did not have any role in this manuscript and we have included this statement in the revised manuscript. 

Round 2

Reviewer 1 Report

Accepted

Reviewer 2 Report

The revised version is improved. I accept it in its present form. Take care of the journal format guidelines. Thank you !i!